# Concurrent Changes in Extreme Hydroclimate Events in the Colorado River Basin

Katrina E. Bennett [1,*], Carl Talsma [1] and Riccardo Boero [2]

1    Los Alamos National Laboratory, Earth and Environmental Sciences, Los Alamos, NM 87545, USA; talsmac83@lanl.gov
2    Los Alamos National Laboratory, Analytics, Intelligence and Technology, Los Alamos, NM 87545, USA; riccardo@lanl.gov
*    Correspondence: kbennett@lanl.gov

**Abstract:** Extreme events resulting in catastrophic damage have more than doubled in the last five years, costing hundreds of lives and thousands of homes, and heavily undermining regional economic stability. At present, most of these hydroclimatic extreme events are documented by the media as individual events; however, in scientific terms, many are better understood as concurrent events—concurrent extremes of both temperature and precipitation (e.g., drought, floods). This paper considers concurrent changes in hydroclimate extremes, including heatwaves, drought, flooding, and low flows, in six historical-to-future (1970–1999, 2070–2099) Earth System Model (ESM) climate scenarios for the Colorado River basin. Results indicate that temperature-driven *Impacts* (heatwaves, drought) have the strongest responses while percipitation-driven *Impacts* have weaker responses. All *Impacts* exhibit an increase in magnitude from synoptic to annual time scales, with heatwaves increasing in strength about three times at the annual time scale versus the synoptic, while low flows only increase slightly. Critical watersheds in the Colorado were identified, highlighting the Blue River basin, Uncompahgre, East Taylor, Salt/Verde watersheds, locations of important water infrastructures, water resources, and hydrological research. Our results indicate that concurrent extreme hydroclimate events are projected to increase in the future and intensify within critical regions of the Colorado River basin. Considering extreme hydroclimate events concurrently is an important step towards linking economic and social effects of these events and their associated instabilities on a regional scale.

**Keywords:** extreme events; hydrology; concurrent; climate change; Colorado River basin; heatwaves; drought; flooding; low flows



## 1. Introduction

Extreme events resulting in catastrophic damage (i.e., loss of life and costs exceeding a billion dollars in response expenses, property damage, and economic disruption) have more than doubled in the last five years, significantly undermining regional economic stability [1]. The increase in both the frequency and intensity of these extreme events has been directly linked to climate change (i.e., 6 events in a 2011 publication, versus 28 events in more recent work just four years later) [2–7]. These extreme events occur locally, regionally, and globally, with major consequences for every aspect of human society and economy [3,7].

The growing body of research on this topic reflects the fact that it has become one of the most urgent issues of the last decade. Studies with the keywords "extreme events" in their titles doubled between 2010 and 2018 (Google Scholar and Scopus, 26 June 2020, Figure 1), and, in a recent study on the "Twenty three unsolved problems in hydrology," "extreme events" was listed as a major outstanding research topic, with particular emphasis placed on the causes of flood/drought periods and recently documented changes to these periods [8]. Extreme events were also the focus of a 2012 *Intergovernmental Panel on Climate Change*

*Special Report* [7], an updated 2018 *Special Report* [3], a "Key Finding" issued by the US National Climate Assessment (NCA 2014), numerous databases and data sets (e.g., NOAA's Billion-Dollar Weather and Climate Disasters https://www.ncdc.noaa.gov/billions/time-series, accessed on 20 October 2020), and a multitude of recent research papers [9–11]. We urgently need to understand how extremes events are changing and we need to better characterize and track such changes in critical regions, such as major watersheds.

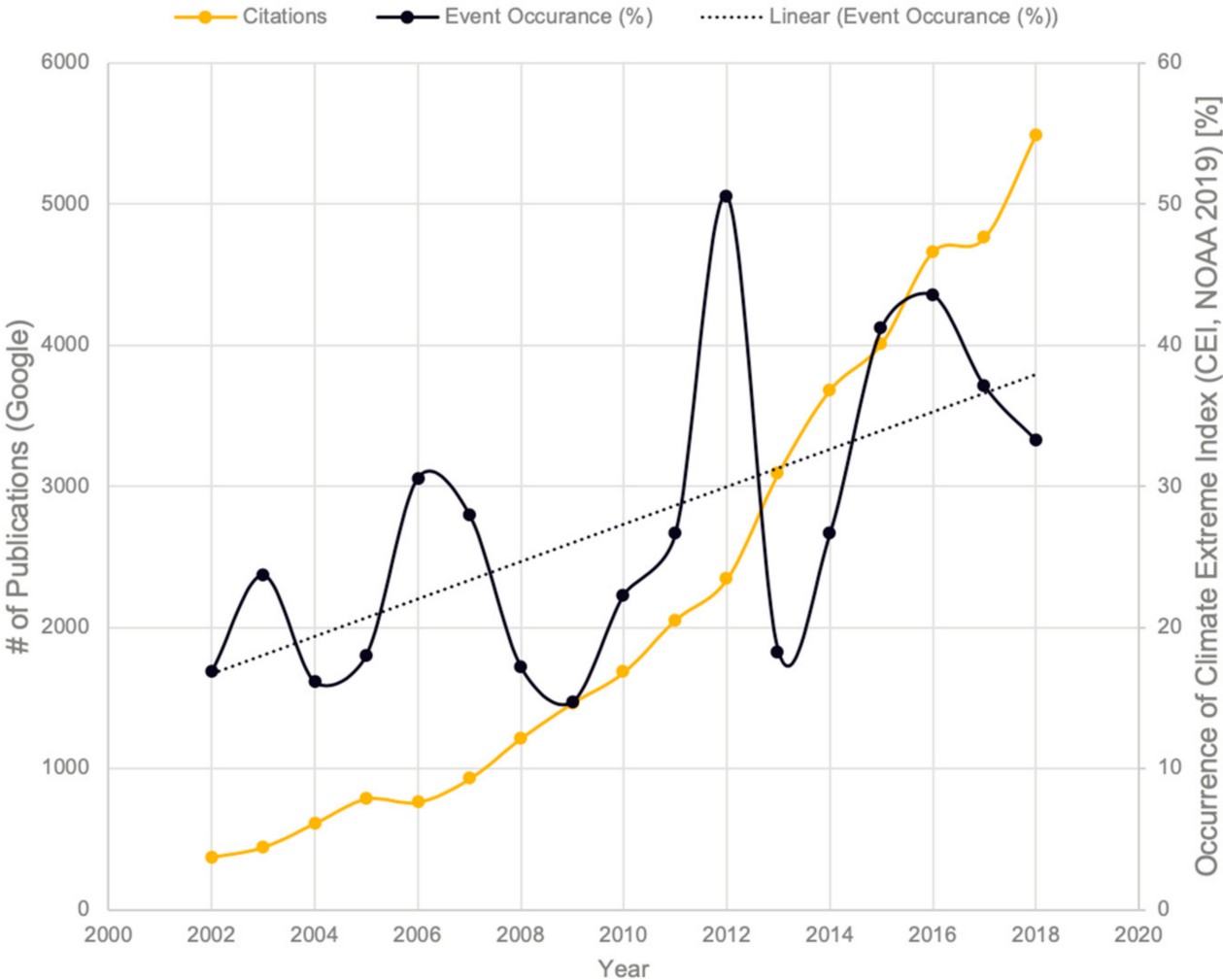

**Figure 1.** Peer reviewed literature documenting extremes events has tracked increases in extremes over the past ~10 years. Data source: NOAA CEI database, 2019.

In the United States, watersheds play a fundamental role in regulating water resources for commercial and domestic purposes. For example, the Colorado River basin (CRB) provides water for over 40 million people [12] and directly facilitates 1.4 trillion dollars in agricultural and commercial applications (roughly 1/13th of the entire US economy, based on the 2014 Gross Domestic Product). However, in the Southwest Climate region alone (UT, CO, AZ, NM), flooding, drought, freezing events, wildfire, severe storms, and winter storms have cost approximately 40 billion dollars between 1980–2020, with a more than 5-fold increase in extreme events from the 1980s to the 2010s [1].

Unfortunately, despite the vital importance of understanding extreme hydroclimate events in watersheds such as the CRB, several fundamental steps have been overlooked. Even the most basic task of comprehensively defining, classifying, and categorizing extreme events has proven to be problematic, complicating comparisons across research fields, which may have different standards for describing the occurrence, duration, and severity of such events [13,14]. Additionally, many of the current tools used to consider

extreme events use simple algorithms (e.g., NOAA's Climate Extreme Index, or CEI) based on simple indicators (minimum temperature changes, precipitation days, etc.), which allow policy makers and the non-scientific community to understand basic changes in extreme event trends. However, these resources cannot effectively track, for example, coupled extreme events (e.g., drought/wildfire, winter storm/flood), and may even overestimate the occurrences of extremes that impact society and underestimate the impacts of concurrent extremes. Consequently, much of the current work on extreme events has focused largely on the analysis of individual or univariate indicators in documenting changing extremes [15–18]. However, more complex indicators and statistical approaches are needed to improve the scientific community's ability to identify and characterize coupled, joint, concurrent, or multivariate climate extremes [19–26] and to document the clear effects of these concurrent events on the economy and society [27,28].

Despite the strong focus on individual indicators, research has emerged that characterizes future changes in extreme events concurrently [22,26,29–38]. Review studies on concurrent hydroclimate extremes provide an assessment of all studies and methodological approaches to date of publication [39,40]. A Special Issue on concurrent or compound extremes was also published [41]. The argument for consideration of concurrent extremes over univariable extreme event analysis is given by several authors, offering definitions, frameworks, paradigms shifts, and generally making the case that examining extreme events in this manner is a means to improve projections of future changes, and that rethinking the traditional univariate approach will allow for physical sciences to be more clearly linked to socioeconomic impacts of extremes [42–46].

This paper describes modeled scenarios of concurrent extreme events in the CRB. We present the results for six different Earth System Models (ESMs), and four different *Impacts* (heat waves, drought, low flows, and flooding), calculated as coupled, joint, multivariate, or concurrent indicators for two time periods, historical (1970–1999) and future (2070–2099). For the purposes of this paper, extreme *Impacts* are considered as the 95th percentile exceedance of concurrent indicators (from here on in, joint, coupled, multivariate indicators are referred to as concurrent). We consider the changes in *Impacts* across synoptic, monthly, seasonal, and annual time scales; and we also identify the most critical watersheds within the CRB where concurrent changes are compounded, based on the accumulated changes in *Impacts* between historical and future models. We conclude the paper with a summary of the work and next steps.

## 2. Materials and Methods

### 2.1. Study Site

Our study area is the CRB, covering an area of 640,000 km$^2$ in the semi-arid to arid Southwestern United States and Northern Mexico. The basin stretches from 30° N to 44° N and from 106° W to 116° W, extending from the alpine regions of the Southern Rocky Mountains to the Gulf of California, and covers elevations from sea level to more than 4000 m, providing water resources to a vast number of adjacent infrastructures (e.g., agricultural, water routing structures, and cities, Figure 2). The large range of spatially and temporally variable CRB landscapes and ecosystems span multiple climatic zones, with observed annual average temperatures ranging from 4 °C to 24 °C (average 11 °C) and annual average precipitation total ranging from 79 mm to 1699 mm (average 363 mm) [47]. Most precipitation in the basin falls as snow at high elevations, so roughly 85% of the CRB flow originates between its upper headwaters and Lees Ferry at Glen Canyon, Arizona [48].

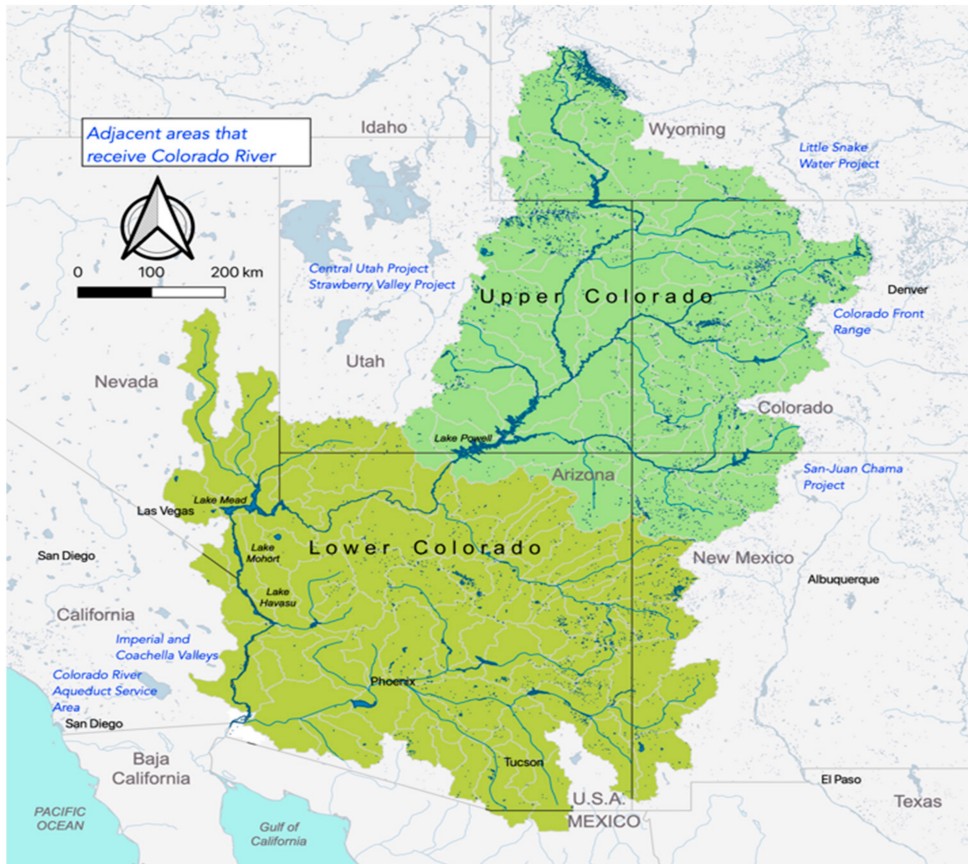

**Figure 2.** The domain of the Colorado River Basin with adjacent areas that receive Colorado River water. Adapted from USGS (accessed 11th January 2021)

*2.2. Climate Simulations*

Our hydrological model was forced using downscaled projections of daily temperature, precipitation, and wind speed from the Multivariate Adaptive Constructed Analogue (MACA) database [49]. We extracted the following six climate data projections from MACA based on Earth System Model (ESM) simulations from the Coupled Model Intercomparison Project, phase 5 (CMIP5) [50] because they represent a range of future climate responses for the CRB and also include dynamic vegetation components, listed in the brackets below: HadGEM2-ES (TRIFFID) [51,52], MIROC-ESM (SIEB-DGVM) [53,54], MPI-ESM-LR (JSBACH) [55,56], IPSL-CM5A-LR (ORCHIDEE) [57,58], and GFDL-ESM2M, and GFDL-ESM2G (LM3V), [59,60]. Each ESM differs by a multitude of factors, including but not limited to the country where the model was developed, initial conditions, physics representations, and tuning mechanisms [61]. For this work, we used the representative concentration pathway (RCP) 8.5 emissions scenario, which tracks closely with changing emissions levels over time [62] and anticipates strongly increasing emissions by 2100 [63].

*2.3. Hydrological Simulations*

To generate simulations of extreme hydrological indicators, we used the Variable Infiltration Capacity (VIC) model version 4.2 [64,65] at a 1/16th degree (6 km) spatial resolution. For each grid cell in the simulation domain, VIC simulates vertical energy and water dynamics at an hourly time step for land cover tiles situated above a 3-layer soil column. Heterogeneity in VIC infiltration is represented by a sub-grid scale statistical distribution (the variable infiltration capacity curve). Surface runoff is generated via saturation excess, while sub-surface runoff is characterized by a non-linear baseflow curve [66].

We ran VIC using publicly available gridded historical climate data (daily precipitation, minimum and maximum temperature, and wind speed) for the CRB (Livneh et al., 2015) and

calibrated VIC for 2006–2010 using the United States Geological Survey (USGS) naturalized gauged monthly streamflow data [67] and an automatic calibration tool [68] to correct streamflow peaks, volume, and low flow biases. See Bennett, et al. [69] for a complete description of VIC parameterizations and calibration details.

### 2.4. Extreme Indicators and Impacts

We calculated two time periods for our analysis: historical (1970–1999) and future (2070–2099) for synoptic (5 days, with 73 5-day intervals in each year), monthly (12-months), four-seasonal (December-January-February, DJF; March-April-May, MAM; June-June-August, JJA; and September-October-November, SON), and annual intervals. Leap-year days were removed from the time series, so all modeled years are 365 days. We calculated extreme climatic and hydrologic indicators (e.g., maximum temperature, freezing days, minimum streamflow) for each time period and interval, as described in Table 1. Some indicators were VIC model input (e.g., temperature) while others were generated from VIC model output (e.g., evapotranspiration). Our defined *Impacts* were: heatwaves, drought, flooding, and low flows. For each *Impact*, we choose two to three indicators that showed the least correlation with one another, yet remained relevant to the assigned *Impact* (Table 2).

**Table 1.** Extreme indicators, description and units, and abbreviations used in the text.

| Indicators | Description and Units | Abbreviation |
|---|---|---|
| Maximum temperature | Maximum temperature achieved over the time period (°C) | *tx* |
| Maximum precipitation | Maximum daily precipitation over the time period (mm) | *precx* |
| Low precipitation days | Number of days when accumulated precipitation is <0.01 mm. (count) | *dryd* |
| Maximum streamflow | Maximum daily streamflow over the time period (mm) | *qx* |
| Minimum streamflow | Minimum daily streamflow over the time period(mm) | *qn* |
| Maximum soil moisture | Maximum daily soil moisture from the third soil moisture layer over the time period (mm) | *soilmx* |
| Minimum soil moisture | Minimum daily soil moisture from the third soil moisture layer over the time period (mm) | *soilmn* |
| Maximum evapotranspiration (ET) | Maximum daily evapotranspiration over the time period (mm) | *evapx* |

**Table 2.** Extreme impacts and the relevant indicators used to construct the impact.

| Impacts | Indicators |
|---|---|
| Heat Waves | Maximum temperature, maximum ET |
| Drought | Maximum temperature, low precipitation days, minimum soil moisture |
| Low Flows | Minimum streamflow, minimum soil moisture, maximum ET |
| Flooding | Maximum precipitation, maximum streamflow, maximum soil moisture |

#### 2.4.1. Peaks Over Threshold Extreme Exceedance

We defined the occurrence of extreme indicator values by calculating the historical 95th percentile (or 5th percentile for low extreme indicators) at each grid cell location for each indicator and using the 95th percentile as the threshold. Following a Peaks Over Threshold (POT) approach, any instance where an indicator value exceeded the historical 95th percentile threshold for a given grid cell was considered extreme. The exceedance of the threshold for each indicator at each grid cell and timestep was expressed as a binary value and is referred to as "exceedance" from here on.

#### 2.4.2. Distance Number

Distance Number $(D_n)$ is a metric that indicates the distance between data pairs normalized by the variance of the observed or historical values. By normalizing the difference by the historical standard deviation, we can obtain a value analogous to a Z-

score, describing the change that occurs between historical and future data [70–72]. We define the Distance Number as:

$$Dn_{xy} = \frac{1}{n} \sum_{i=1}^{n} \frac{(F_{i,xy} - H_{i,xy})}{S_{xy}}, \tag{1}$$

where $F$ is the future indicator exceedance, $H$ is the historical indicator exceedance, $S$ is the temporal standard deviation of the historical exceedance for each cell, $i$ is the timestep, $xy$ is the spatial cell location, and n is the number of timesteps. We calculated the distance number using equation (1) and estimate the change in exceedance of extreme *Impacts* by averaging $D_n$ for each indicator relevant to a given *Impact*. For example, the concurrent $D_n$ value for flooding is the average $D_n$ value of the $D_n$ of maximum precipitation, maximum streamflow, and maximum soil moisture, respectively. By normalizing the exceedance values by the historical standard deviation, the concurrent $D_n$ value allows for a comparison of change across large spatial distances and four different time scales, and also allows for a comparison between different *Impacts* and ESMs. $D_n$ values were examined for individual grid cell responses, and by the Natural Resource Counsel (NRCS), United States Department of Agriculture (USDA) hydrological unit basins. We used the cataloguing unit level eight (HUC8) watersheds, of which there are 134 watersheds in the CRB.

We estimated critical $D_n$ values based on the mean of the $D_n$ for each grid cells in all 134 HUC8 watershed in the CRB. The HUC8 watersheds where the mean $D_n$ exceeded the 95th percentile of the mean $D_n$ were identified as critical for all *Impacts* and ESMs. For all HUC8s, we summed instances for *Impacts* and ESMs; watersheds with values above 4 are identified as notably critical (described as having criticality and compounded concurrent extremes in Results and Discussion sections).

## 3. Results

The six ESMs examined in this study exhibit very similar temperature and precipitation data across the historical time period. Each ESM exhibits a mean temperature across the entire CRB of between 11.7 °C and 11.9 °C, and an average annual precipitation between 363.5 mm/yr and 369.9 mm/yr (*std. dev.* = 6.4 mm/yr, Table 3). The ESMs were chosen to represent the range of changes in precipitation and temperature projected by the larger CMIP5 model suite (not shown). Among the six ESMs, the projected change in temperature for the future scenario (2070–2099) differenced from the historical scenario (1970–1999) ranges from an increase of 4.1 °C (GFDL-ESM2M) to 7.0 °C (MIROC-ESM), with a multi-model average of 5.5 °C (*std. dev.* = 1.2 °C). Projections of future precipitation show large variance among ESMs, with ESMs projecting both increases and decreases in annual precipitation. HadGEM2-ES365 shows the largest decrease in precipitation (−52.8 mm/yr) while GFDL-ESM2G shows the largest increase (37.2 mm/yr). The mean of the six ESMs shows a small increase of only 2.1 mm/yr in precipitation and a standard deviation between ESMs of 32.6 mm/yr.

**Table 3.** Future and historical average annual temperature and precipitation values from the six GCMs considered. We also include the multi-model average and standard deviation for each field (bold italics). Temp. = temperature, Prec. = precipitation.

|  | **Historical** | | **Future** | | **Change** | |
| --- | --- | --- | --- | --- | --- | --- |
|  | **Temp. (°C)** | **Precip. (mm)** | **Temp. (°C)** | **Precip. (mm)** | **Temp. (°C)** | **Precip. (mm)** |
| GFDL-ESM2G | 11.7 | 365.9 | 16.3 | 403 | 4.5 | 37.2 |
| GFDL-ESM2M | 11.7 | 366.5 | 15.8 | 377.9 | 4.1 | 11.3 |
| HadGEM2-ES365 | 11.8 | 366.8 | 18 | 360.4 | 6.2 | −6.4 |
| IPSL-CM5A-LR | 11.9 | 351.7 | 18.2 | 298.9 | 6.3 | −52.8 |
| MIROC-ESM | 11.8 | 369.9 | 18.8 | 400.7 | 7 | 30.8 |
| MPI-ESM-LR | 11.9 | 363.5 | 17 | 356.3 | 5.1 | −7.3 |
| Average | **11.8** | **364.1** | **17.3** | **366.2** | **5.5** | **2.1** |
| Standard Deviation | **0.1** | **6.4** | **1.2** | **38.3** | **1.2** | **32.6** |

### 3.1. Individual Indicators

Figure 3 shows the average across all six ESMs and four time scales for the individual indicators contributing to each *Impact*. Maximum temperature shows the largest $D_n$ values of any indicator, exhibiting an average of 2.07 across each ESM and time scale. Additionally, maximum temperature (*tx*) shows little spatial variance (*std. dev.* = 0.07), which is not surprising given that temperature often exhibits coherent and strong responses under climate change [73]. Maximum precipitation (*precx*) shows little spatial variance (*std. dev.* = 0.09) with modestly positive $D_n$ values (0.22) across the entire basin, indicating an increase in maximum precipitation for the CRB as a whole. $D_n$ values are slightly higher in the upper CRB than in the lower basin for maximum precipitation. Similarly, maximum evaporation (*evapx*) and dry days (*dryd*) show slightly positive values (0.16, and 0.14, respectively) across nearly the entire CRB with little spatial variability (*std. dev.* = 0.08 to 0.13). Maximum evaporation shows slightly more elevated $D_n$ values in the highest elevations regions of the CRB, as well as in a narrow band in the lower CRB known as the Arizona Transition Zone, where topography transitions from the high elevation Colorado Plateau to the basin and range region of the US southwest [74,75]. This region is also characterized as experiencing the highest average annual rainfall in the state of Arizona [76].

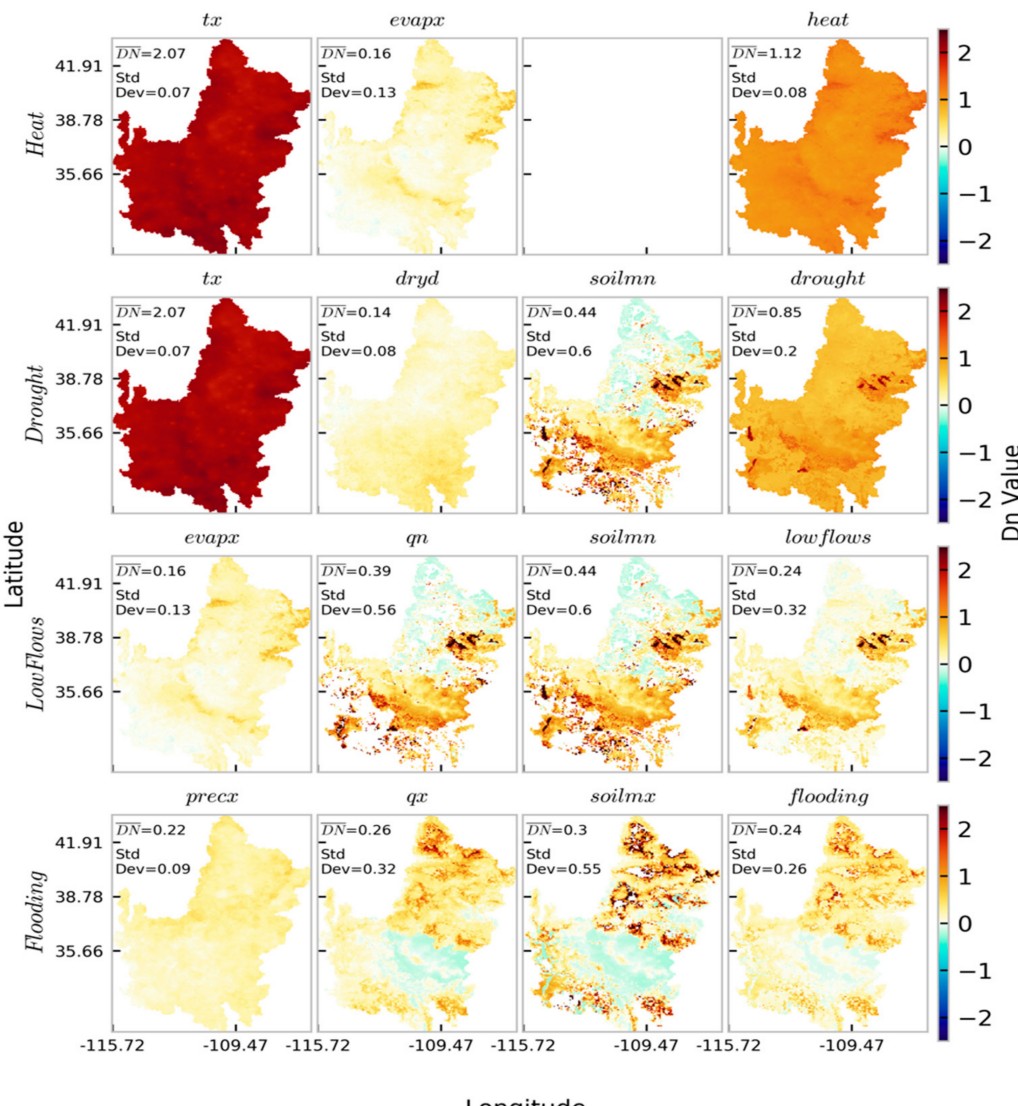

**Figure 3.** The multi-model and timescale average of $D_n$ results for the individual indicators along with the relevant impacts.

Indicators related to minimum (*std. dev.*= 0.56) or maximum streamflow (*std. dev.*= 0.32), or minimum (*std. dev.*= 0.60) or maximum soil moisture (*std. dev.*= 0.55) exhibit far more spatial variance than the other indicators (*qn*, *qx*, *soilmn*, soilmx, respectively). Maximum soil moisture and maximum streamflow have larger and more positive $D_n$ values in the upper CRB compared to the lower CRB, while minimum soil moisture and maximum soil moisture show larger and more positive $D_n$ values in the lower CRB compared to the upper CRB.

### 3.2. Impacts

$D_n$ results as difference between the future scenario and historical scenario for each *Impact* and time scale are shown in Figures 4–8 and Table 4. Overall, we observe an increase in each of the four *Impacts* when averaged over all ESMs and across the entire CRB (Table 3, Figure 4). Generally, the $D_n$ values increase as the time scale lengthens, with the largest values occurring on the annual time scale (Figure 4). However, the change in $D_n$ values depends on the *Impact*. Temperature-driven *Impacts* (heatwaves, drought) exhibit an increase in magnitude from synoptic to annual time scales, with heatwaves resulting in approximately three times the $D_n$ value at the annual time scale versus the synoptic (0.60 compared to 2.01, Table 3, Figure 4). On the other hand, precipitation-driven *Impacts* (low flows, flooding) exhibit only a slight increase when moving from synoptic to annual time scales (e.g., 0.21 compared to 0.32 for low flows, Table 3, Figure 4). This may be because precipitation-driven extremes have a shorter memory in the climate system than temperature-driven extremes, which often occur over longer periods of time and have longer memories in terms of both reoccurrence and frequency [77].

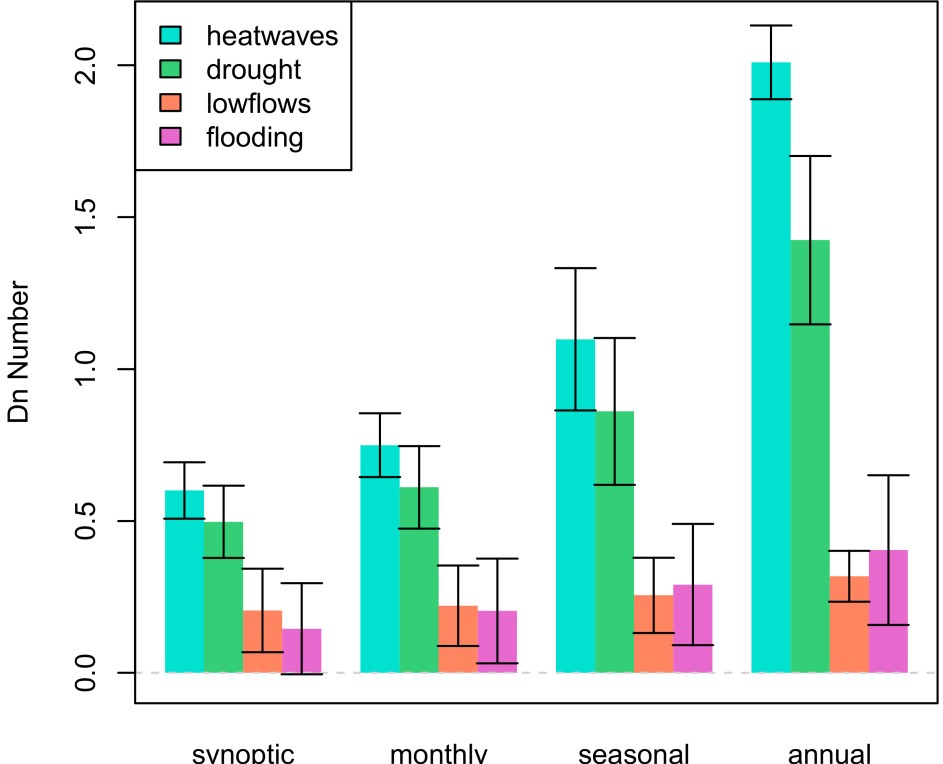

**Figure 4.** The multi-model average $D_n$ value across the CRB for each impact and timescale. The whiskers represent +/− one standard deviation between the different ESMs.

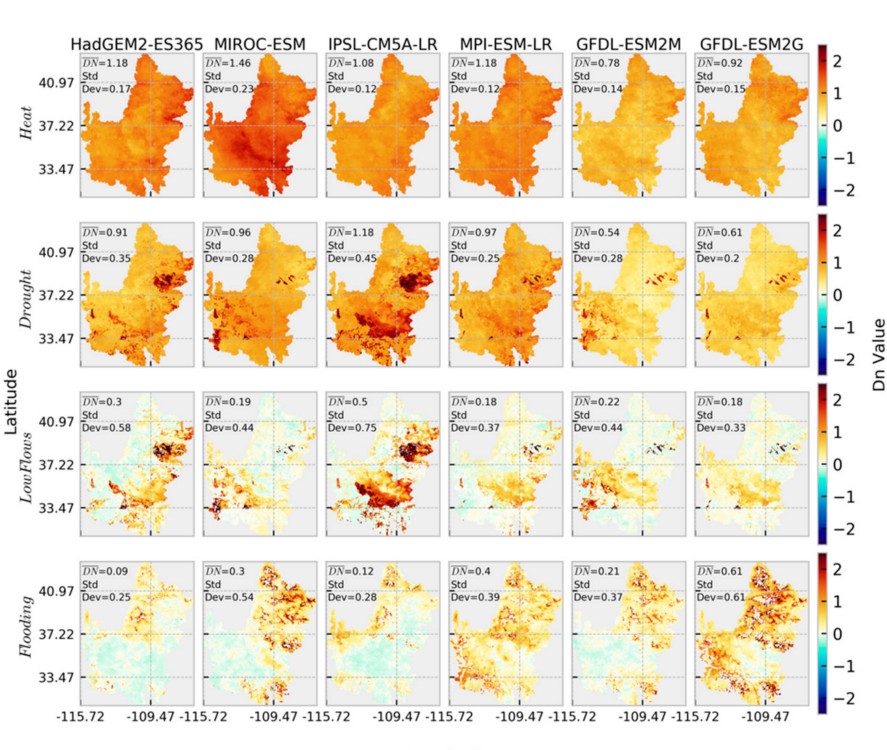

**Figure 5.** The $D_n$ value results across the CRB at an annual timescale for each ESM and Impact. The average $D_n$ value and spatial standard deviation is also shown for each panel.

**Figure 6.** The $D_n$ value results across the CRB at a seasonal timescale for each ESM and Impact. The average $D_n$ value and spatial standard deviation is also shown for each panel.

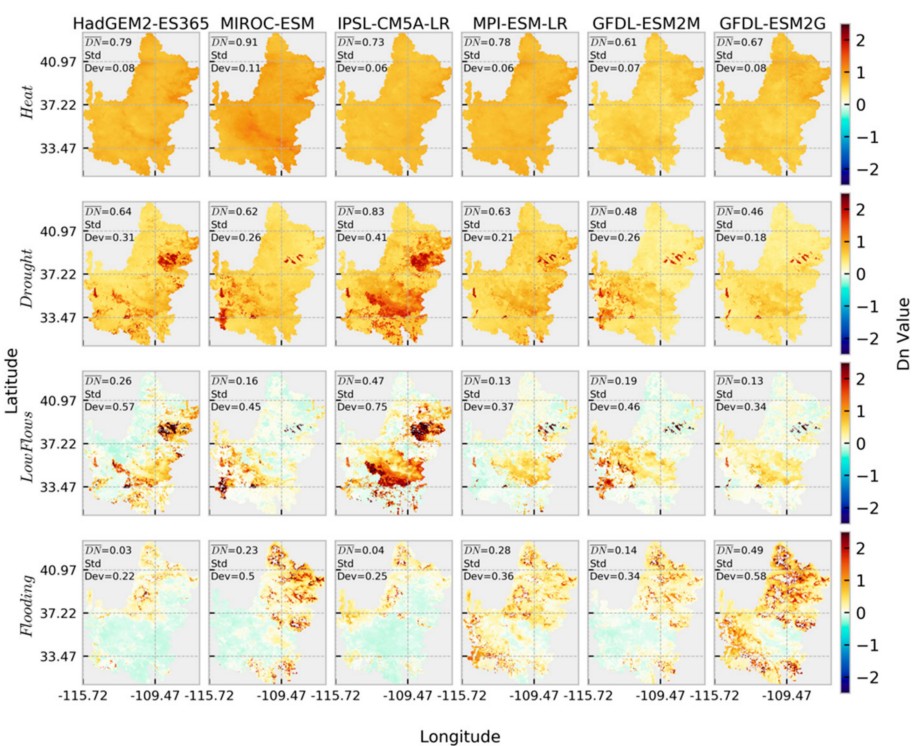

**Figure 7.** The $D_n$ value results across the CRB at a monthly timescale for each ESM and Impact. The average $D_n$ value and spatial standard deviation is also shown for each panel.

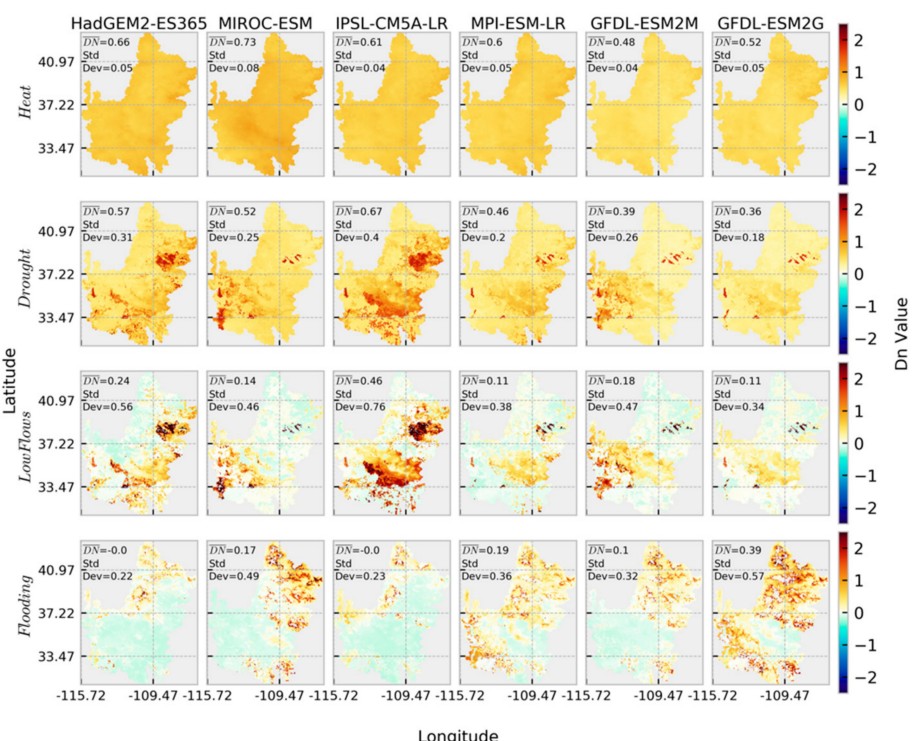

**Figure 8.** The $D_n$ value results across the CRB at a synoptic timescale for each ESM and Impact. The average $D_n$ value and spatial standard deviation is also shown for each panel.

**Table 4.** Multi-Model Average $D_n$ values for each impact and timescale. The table includes $D_n$ values for the entire CRB, Upper CRB, and Lower CRB as well as the standard deviation between GCMs in parenthesis.

|           |            | Synoptic    | Monthly     | Seasonal    | Annual      |
|-----------|------------|-------------|-------------|-------------|-------------|
| CRB       | Heatwaves  | 0.60 (0.09) | 0.75 (0.11) | 1.10 (0.23) | 2.01 (0.12) |
|           | Drought    | 0.50 (0.12) | 0.61 (0.14) | 0.86 (0.24) | 1.42 (0.28) |
|           | Low Flows  | 0.21 (0.14) | 0.22 (0.13) | 0.26 (0.12) | 0.32 (0.08) |
|           | Flooding   | 0.14 (0.15) | 0.20 (0.17) | 0.29 (0.20) | 0.40 (0.25) |
| Upper CRB | Heatwaves  | 0.60 (0.08) | 0.75 (0.10) | 1.11 (0.19) | 2.02 (0.13) |
|           | Drought    | 0.44 (0.13) | 0.55 (0.15) | 0.79 (0.25) | 1.35 (0.29) |
|           | Low Flows  | 0.14 (0.15) | 0.15 (0.15) | 0.19 (0.15) | 0.27 (0.14) |
|           | Flooding   | 0.27 (0.20) | 0.34 (0.22) | 0.44 (0.24) | 0.60 (0.29) |
| Lower CRB | Heatwaves  | 0.60 (0.10) | 0.75 (0.12) | 1.09 (0.27) | 2.00 (0.14) |
|           | Drought    | 0.55 (0.12) | 0.66 (0.14) | 0.92 (0.25) | 1.49 (0.28) |
|           | Low Flows  | 0.27 (0.15) | 0.28 (0.14) | 0.31 (0.13) | 0.36 (0.08) |
|           | Flooding   | 0.04 (0.12) | 0.09 (0.15) | 0.16 (0.18) | 0.24 (0.23) |

Figures 5–8 show the spatial detail of $D_n$ value results across the CRB for each of the *Impacts*, ESMs, and time scales. The $D_n$ values show strong and spatially coherent increases in both heatwaves and drought across the entire basin for each of the four time scales. As temperature increases is occurring across the CRB, maximum temperature presumably drives many of the spatially consistent increases for both drought and heatwaves [73]. MIROC-ESM, the ESM with the warmest projected temperature across the basin, also showed the largest *Impacts* and $D_n$ values of the six ESMs (2.18, Figure 5 and repeated in Figures 6–8).

The $D_n$ values of drought often exhibit more spatial variability than heatwaves. Of the six ESMs, IPSL-CM5A-LR is the driest of the ESMs (1.88, Figure 5 but repeated in Figures 6–8), generally exhibiting the highest $D_n$ values and showing especially severe drought in the Little Colorado River basin and in areas along the southern half of the CRB and the Colorado headwaters (e.g., Figure 5, column 3). Of the indicators, minimum soil moisture generally exhibited the largest $D_n$ values in these areas as well, and was the most spatially variable indicator contributing to drought (see Figure 3). Both of the GFDL ESMs exhibited the lowest $D_n$ values for drought, likely due to the more modest temperature increases exhibited by these ESMs (1.19 and 1.28, Figure 5, Table 3).

In contrast, low flows and flooding exhibited both positive and negative values throughout the CRB, although the entirety of the CRB, when averaged, exhibited positive $D_n$ values for both *Impacts* at all time scales (Figures 5–8). Flooding is the stronger indicator at annual time scales in the entire CRB, but low flows have higher $D_n$ values at the synoptic scales, and both indicators have a similar magnitude of response at monthly and seasonal scales (Table 4). Agreement across ESMs illustrates higher coherency (lower variances) in low flows at longer time scales, while lower coherency (higher variance) is shown at longer time scales for flooding. Figures 5–8 show the spatial variations in responses across ESMs, with the driest ESM having the strongest responses for low flows (0.50, IPSL-CM5A-LR), and the wettest ESM (0.80, GFDL-ESM-2G) having the strongest responses for flooding.

The high-elevation, snow-dominated upper CRB and the low-elevation, arid lower CRB respond differently to climate change effects, and this dichotomy carries through the $D_n$ results for *Impacts* and time scales for the Upper and Lower CRB (Table 4). Heatwaves show very little difference between Upper and Lower CRB (0.00 to 0.03) and generally have very little spatial variance, as discussed above. Both drought (−0.11 to −0.14) and low flows (−0.10 to −0.13) exhibited lower $D_n$ values in the Upper CRB than in the Lower CRB. Flooding exhibits the greatest difference between Upper and Lower basins (0.23 to 0.37), with larger $D_n$ values in the Upper CRB. Overall, the lowest $D_n$ values are flooding in the Lower CRB at synoptic time scales (0.04), while low flows is lowest in the Upper CRB at synoptic time scales (0.14).

To account for the regions of the CRB where cumulative concurrent changes are occurring, we summed the instances when the *Impacts* or ESMs for HUC8 watersheds were greater than the 95th percentile of mean $D_n$ value (Figure 9). The watersheds with the highest count of these instances are the Blue River basin (10), followed by Little Colorado River Headwaters (7). After these two HUC8 watersheds, the East-Taylor, Uncompahgre, San Miguel, and Big Chino-Williamson Valley basins each have six instances of critical concurrent extreme *Impacts*. Gunnison, Dolores, Verde, Lower Salt, and all the other watersheds identified in Figure 9 not already discussed have 4 concurrent extreme *Impacts*. The Uncompahgre had the most instances of criticality for low flows (4), followed by the adjacent San Miguel basin (3). San Miguel also had the greatest number of instances of drought of any watershed (3). Across the *Impacts* heatwaves has the most instances of criticality, with Blue River having the greatest (5), followed by the Little Colorado River Headwaters, East-Taylor, and the Big Chino-Williamson Valley watersheds (3 for each HUC8).

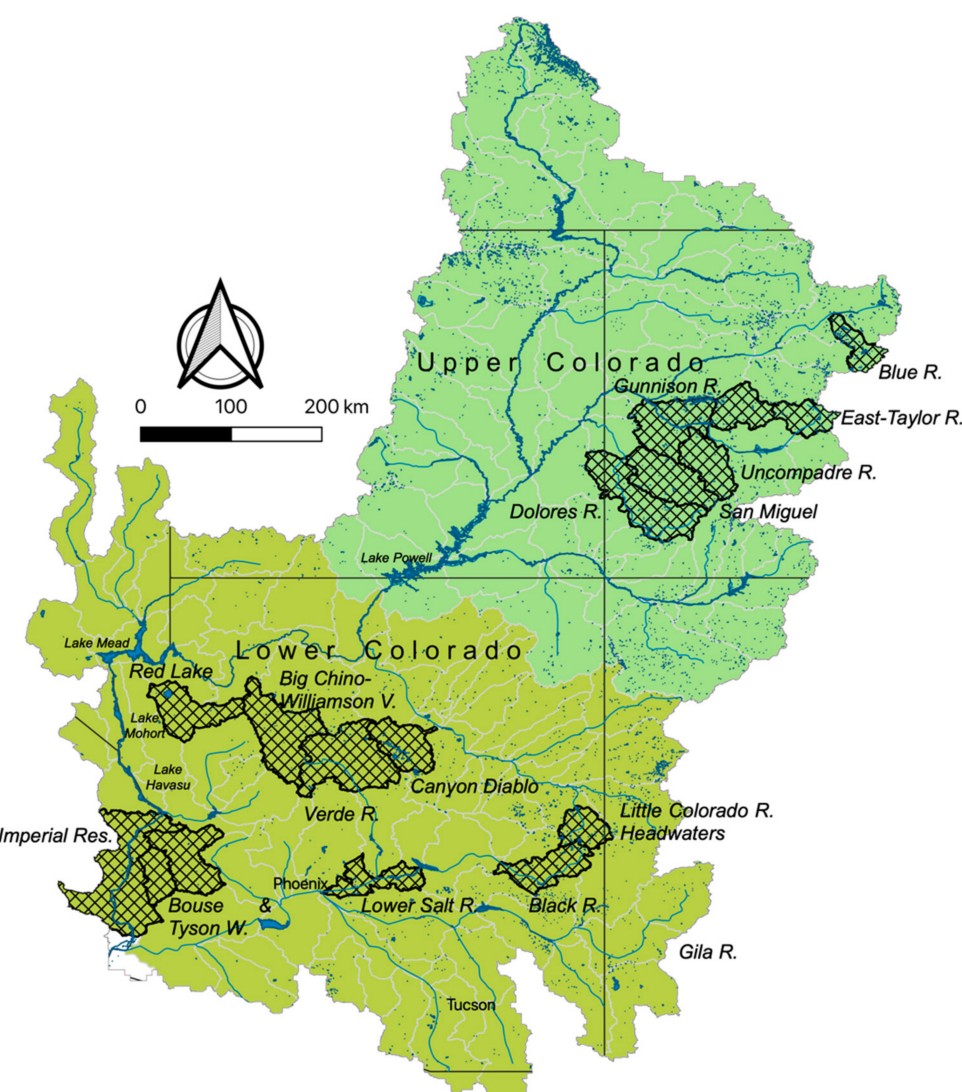

**Figure 9.** Critical basins in the Colorado River basin.

## 4. Discussion

The projected changes in temperature in the CRB over the historical to future time periods are drastic, but not unexpected. The average increase in temperature reflects estimates of regional change over the same period [78]. While there is some uncertainty between the ESMs in the magnitude of temperature increase, it is relatively small compared to the

uncertainty in the ESM's estimates of future precipitation, a result that is also found across many studies of climate change [79]. Unlike estimated changes in temperature, estimated changes in precipitation vary greatly between the ESMs, and there is no agreement on the direction of change, with an almost equal average annual precipitation *decrease* modeled by IPSL-CM5A-LR and an annual precipitation *increase* modeled by GFDL-ESM2G. However, it should be noted that a wet ESM projection for precipitation may not lead to a less arid environment in the CRB, as potential increases in evaporation due to air temperature increases could skew the region towards greater aridity despite precipitation increases [80].

Despite the wide range of climate projections from a range of ESMs, the $D_n$ values show (with a few minor exceptions) a remarkably similar spatial pattern for the extremes that include temperature indicators, such as heatwaves and drought. These *Impacts* also have the strongest signal of extremes across the CRB [81]. Therefore, heatwaves and drought will likely affect the CRB on a uniform basis and most strongly, with some pockets of higher and lower responses related to antecedent moisture conditions owing to drier or wetter soils in the case of droughts [82].

On the other hand, the spatial pattern for low flows and flooding is more variable across the CRB, owing to their spatially varying drivers and indicators such as precipitation, streamflow, and soil moisture. For both *Impacts*, soil moisture (minimums or maximums for low flows or flooding, respectively) had the strongest response among the indicators, illustrating the importance of soil moisture on these extremes. The effect of antecedent soil moisture on flooding is well documented in studies of historical events, forecasting, and climate change [83–85]. Although studies on low flows appear less common, Castillo, et al. [86] suggest that antecedent soil moisture is an important control on runoff, particularly during medium- and low-intensity storms, which are common in semi-arid locations such as the Lower CRB. These *Impacts* were thus dependent upon the intensity of those indicators in time within particular regions of the CRB, with notable differences between the Upper and the Lower CRB, indicating the need to examine these *Impacts* within specific regions or watersheds.

In general, the time scale of responses for concurrent extreme events was strongest for $D_n$ values at the annual time scale. *Impacts* such as heatwaves and drought more than doubled from synoptic to annual time scales. This may be because temperature-driven extremes, which often occur over longer periods of time have longer memories in climate systems in terms of both reoccurrence and frequency [77]. Consequently, researchers looking to examine these temperature-driven extremes, the annual time scale is perhaps the most appropriate scale at which to examine these *Impacts*. Although results for low flows and flooding were also strongest at the annual time scale, these *Impacts* had much less change in their response across time scales across the CRB, suggesting that the time scale of analysis for these may be aligned with research drivers and/or science questions at hand. For example, a forecasting research method should examine flooding *Impacts* at a synoptic time scale, since forecasting is most concerned with synoptic scale changes in extremes. Additionally, important was the fact that flooding and low flows, also had the greatest disagreement across ESMs at annual time scales.

The Upper CRB and Lower CRB responded differently to concurrent extreme events for both spatial patterns and time scales. Within the Upper CRB, the coherency of low flows did not shift much across time scales, while, for the Lower CRB, coherency decreased (spatial variability increased). Flooding response was higher and low flows was lower in the Upper CRB, a pattern that was reversed for the Lower CRB, indicating the importance of changing floods for the Upper basin, while low flows are a larger issue for the Lower CRB. This result is not unexpected owing to the moisture deficits regimes of the Lower CRB, and the moisture excess regime of the Upper Colorado [48].

Our findings concur with the previous, limited research on extreme concurrent events, although to our knowledge, studies have not been undertaken specifically for the CRB. For example, univariate conditions for temperature and droughts showed no change, but concurrent droughts and heatwaves were shown to be occurring across the US [22];

a similar study showed that wind speeds were not increasing univariately across the Midwest, hot, dry, and windy events were increasing [87]. Analysis of concurrent extreme events was undertaken in a study by Hao, AghaKouchak and Phillips [26] where they used 13 CMIP5 models to examine concurrent extremes as scenarios, showing that concurrent warm/dry and warm/wet global extreme events have increased gradually since the ~1960s and substantially since the ~1990s, with warm/wet extreme events increasing in high latitudes and tropical regions, and warm/dry extremes increasing in many areas, including central Africa, eastern Australia, northern China, parts of Russia, and the Middle East [26]. Recent extremes in California were examined by Diffenbaugh, Swain and Touma [42], who highlights the role of record high annual mean temperatures in combination with record low annual mean precipitation in 2013, which led to increased evapotranspiration, more intense drought, and intensifying wildfire occurrences. Likewise, studies that consider changes such as forest disturbances under a warming climate are also looking at concurrent extreme events [88], although studies may not necessarily self-identify as examples of concurrent extreme event analysis.

Identifying critical regions within the CRB is important to determine the vulnerability of specific watersheds to concurrent extreme events. The watersheds identified in this work had instances of multiple ESMs and/or multiple *Impacts* projected to occur under the annual interval, a time scale most noted for its implications for water management of the system $\chi$. Examining the entire CRB and being able to rank the most critical watersheds allows us to focus more closely on those systems in terms of resources and research. The critical basins identified in this work are indeed systems that have been well studied and/or have important water resources located within them. For example, Livneh, et al. [89] examined the Uncompahgre River basin in his investigation of forest disturbance and dust-on-snow implications, a study on concurrent events, although not specifically identified as such. The East-Taylor basin was selected as a 'representative community watershed' for intensive study by The Watershed Function project, funded by the USDOE Biological and Environmental Research Subsurface Biogeochemistry Program [90]. Additionally, the Salt/Verde complex has experienced damaging extreme events leading to effects on humans and economy [91–93]. The Blue River watershed is an important system as it includes the largest reservoir in the Upper CRB, and has been identified as threatened by climate change such as increasing temperatures [94].

This paper examines future changes in concurrent extreme events for different time scales in the CRB, focusing largely on a basic statistical metric to combine extremes. Our study is limited in that is leaves out important work to understand which extreme indicators to combine together to represent *Impacts*, and also more advanced statistics metrics, is fundamental to testing the applications and results illustrated herein. The next phase of this work includes a more advanced statistical method using copulas to characterize and describe changing concurrent extremes across the CRB. Under the same project, we have developed an economic model to consider the effects of modeled flooding in the CRB on the US economy under future climate change.

## 5. Conclusions

We analyzed extreme concurrent *Impacts* of heatwaves, drought, low flows, and flooding for the CRB under multiple future (2070–2099) climate change scenarios. Temperature and precipitation are projected to change dramatically in the CRB: all ESMs predict temperature increases, but different ESMs predict either significant increases or significant decreases in precipitation. Despite this range in response, extreme concurrent events are projected to increase across the CRB in all time scales and for all *Impacts.*

Temperature-driven concurrent extremes (heatwaves and drought) are strongest and most spatially coherent across the CRB, while precipitation-driven concurrent extremes (flooding and low flows) are less strong and more spatially variable across the CRB. Annual time scales for analysis of concurrent extremes show the strongest responses for all variables. Temperature-driven concurrent extreme *Impacts* shift the most across the time scales of

analysis, with a more than doubling of response from synoptic to annual time scales. Precipitation-driven concurrent extremes do not show as much of a change across time scales, with a slight increase in response moving from synoptic, monthly, seasonal, to annual time scales. However, there is generally greater agreement moving from the annual to synoptic time scales; with the exception of low flows that has greater ESM coherence at annual time scales.

The Upper and Lower CRB act similarly temperature-driven concurrent extremes, again, but different in their response to changes in precipitation-driven concurrent extremes; with flooding having the strongest response in the Upper CRB, and low flows having a stronger response in the Lower CRB. We identified critical watersheds in the CRB that are projected to experience compounded concurrent extreme events, watersheds that include important water management structures in the Upper CRB, such as the Blue River basin, and where vital water research resources are located, such as the Uncompahgre and the East-Taylor basins. In the Lower CRB, the Salt/Verde basin provide important sources of in-basin surface water flow in an otherwise arid environment, and water for agriculture and hydropower at the Roosevelt Dam [95]. Our work verifies that concurrent extreme events are likely to increase in frequency and magnitude under future climate change [7,96,97].

As the science of extreme events evolves, examining concurrent events will likely be an important step in understanding the changing nature of extremes, capturing and analyzing the largest and most damaging of these events, tying instances of extremes in climate and land surface model simulations to observations, and linking the physical responses to other indicators of instability, such as economy and society.

**Author Contributions:** Conceptualization, K.E.B.; methodology, K.E.B., C.T., and R.B.; validation, C.T.; formal analysis, K.E.B. and C.T.; writing—original draft preparation, K.E.B. and C.T.; writing—review and editing, K.E.B., C.T., and R.B.; visualization, C.T.; project administration, K.E.B.; funding acquisition, K.E.B. All authors have read and agreed to the published version of the manuscript.

**Funding:** This work was funded under the Los Alamos National Laboratory Lab Directed Research and Development (LDRD) Early Career Research program (20180621ECR).

**Institutional Review Board Statement:** Not applicable.

**Informed Consent Statement:** Not applicable.

**Data Availability Statement:** Downscaled CMIP5 climate model projections may be downloaded via the MACA web portal: https://climate.northwestknowledge.net/MACA/ (accessed on 20 October 2020). VIC model may be downloaded via GitHub: https://github.com/UW-Hydro/VIC (accessed on 20 October 2020). Historical VIC forcing data may be obtained from ftp://gdo-dcp.ucllnl.org/pub/dcp/archive/OBS/livneh2014.1_16deg/ (accessed on 20 October 2020). Naturalized streamflow data for the Colorado River basin may be obtained from USBR: https://www.usbr.gov/lc/region/g4000/NaturalFlow/current.html (accessed on 20 October 2020) (U.S. Bureau of Reclamation, 2018). Other model parameter files and model outputs may be obtained by contacting the authors.

**Acknowledgments:** We would like to thank Jorge Urrego-Blanco for his help with statistical measures to formulate concurrent extreme events that became a part of this paper. Thank you to the help of an anonymous reviewer for their suggestions and improvements to the text of this paper. Additionally, thank you to Richard Middleton, who has brought KB to Los Alamos and provided initial support under LDRD DR (20150397DR), which allowed KB to develop the VIC modeling work that our analysis is based on (completed under Los Alamos Director's Postdoctoral Fellowship, 20160654PRD).

**Conflicts of Interest:** The authors declare no conflict of interest. The funders had no role in the design of the study; in the collection, analyses, or interpretation of data; in the writing of the manuscript, or in the decision to publish the results.

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
