# Peer review of "Concurrent Changes in Extreme Hydroclimate Events in the Colorado River Basin"

_water, doi:10.3390/w13070978_

Round 1

Reviewer 1 Report

The current manuscript discusses the hydroclimatic trend in the past and future based on an Earth System Model simulation of the Colorado River Basin. The writing is good and the structure is generally well written. The reviewer had the following comments.

  1. Introduction: From the introduction, the reviewer had the impression that no research has done before utilizing ESM on this topic. If that is true, please highlight this fast in the manuscript. If not, please add more descriptions of past research utilizing ESM on hydroclimate simulations. The current introduction is missing this element. If necessary, you can merge section 2.2 into the introduction.
  2. Section 2.2: Please provide brief descriptions of the ESMs. What are considered/not considered in the models? What are their differences? What are some of the representative applications in the past?
  3. Section 2.3: Some of the representative input data should be displayed here as graphs, such as temperature and rainfall.
  4. Section 2.4: The application of the “extremes” is not well-explained. For example, how were “heat waves” simulated? Did you use maximum temperature and ET for every day?
  5. Results: 1) Why are there underlines below the “degree” sign? 2) Discussions for Figure 3 can be more comprehensive to cover most variables studies in the figure. 3) Results can be divided and discussed according to the historical and future simulations. This is particularly important since multiple future scenarios were utilized.

Author Response

Reviewer #1

The current manuscript discusses the hydroclimatic trend in the past and future based on an Earth System Model simulation of the Colorado River Basin. The writing is good and the structure is generally well written. The reviewer had the following comments.

  1. Introduction: From the introduction, the reviewer had the impression that no research has done before utilizing ESM on this topic. If that is true, please highlight this fast in the manuscript. If not, please add more descriptions of past research utilizing ESM on hydroclimate simulations. The current introduction is missing this element. If necessary, you can merge section 2.2 into the introduction.

Author’s Response to Reviewer #1 Question 1:

We have added a section in the introduction that references, in brief, the works that have been carried out utilizing ESMs and other means to examine concurrent extreme events. Please see the track changed version of the manuscript.

  1. Section 2.2: Please provide brief descriptions of the ESMs. What are considered/not considered in the models? What are their differences? What are some of the representative applications in the past?

Author’s Response to Reviewer #1 Question 2:

We have added a sentence to describe the major differences between ESMs, in general, and a reference to a new tool and a recent paper that discusses these differences and some of the history of the global climate models.  We do not feel that more detail than this is warranted. We have tried to briefly describe all the preprocessing steps that were taken in the paper, with the aim of brevity. A detailed description of the ESMs is given in the references that are provided for each model. We think that describing the very ways that the models differ in detail, or any applications (there are many) will not add much to the paper. We respectfully decline to add answers to the above questions to the paper in regards to this comment. If this is somehow a sticking point for the reviewer, please let us know and we can add a sentence or two on the differences of the ESMs.

  1. Section 2.3: Some of the representative input data should be displayed here as graphs, such as temperature and rainfall.

Author’s Response to Reviewer #1 Question 3:

Bennett et al. 2018 shows the temperature and precipitation results for the input data for one basin in the CRB, which is referred to at the end of this section. Would the reviewer like us to add a similar figure to this paper? Again, we do not think adding this information would be a useful addition to the work. However, again, if the reviewer insists on this addition, we can do this.

  1. Section 2.4: The application of the “extremes” is not well-explained. For example, how were “heat waves” simulated? Did you use maximum temperature and ET for every day?

Author’s Response to Reviewer #1 Question 4:

The description of how we calculated extremes is provided in Table 1 and Table 2, and in section 2.4. For heatwaves, for example, we used maximum temperature and maximum ET for every day (for synoptic), for each of the 12 months, for seasons (each four months period), and for annual. We are unsure how we can describe this better for the reviewer. Could the reviewer provide more specific details with regards to how this is not clear in the section?

  1. Results: 1) Why are there underlines below the “degree” sign? 2) Discussions for Figure 3 can be more comprehensive to cover most variables studies in the figure. 3) Results can be divided and discussed according to the historical and future simulations. This is particularly important since multiple future scenarios were utilized.

Author’s Response to Question 5.1:

We have replaced the underlined degree symbols with regular degree symbols. Please let us know if we have missed any of these typographic errors.

Author’s Response to Reviewer #1 Question 5.2:

We discuss each of the indicators (tx, evapx, precx, dryd, soilmn, soilmx, qn, qx) in Figure 3 in the first two paragraphs of section 3.1. I am wondering if the reviewer missed the discussion of qn, qx, soilmn, and soilmx in the second paragraph. Please let us know if there is an indicator that we missed. We added the shorted indicator names to the second paragraph to make that clearer.

Author’s Response to Reviewer #1 Question 5.3:

We chose to write this paper focusing on the Dn statistic. We feel that this use of this statistic is a novel approach that we have not seen used before. It is also a simple way of explaining the differences between the future and historical data, and it puts all indicators and impacts on the same scale (i.e., normalization as described in the paper) for comparison of the greatest/largest effects. We do not think that adding separate analysis and description of the absolute future and historical results would add much to this paper, except to perhaps repeat previous research and/or confuse the reader. We have opted to keep this out of the paper.

Reviewer 2 Report

The manuscript entitled “Concurrent Changes in Extreme Hydroclimate Events in the Colorado River Basin” by Bennett et al. examines concurrent changes in hydroclimate extremes, including heatwaves, droughts, floods, and lowflows, in six historical-to-future (1970-1999, 2070-2099) Earth System Model (ESM) climate scenarios for the Colorado River basin.

The work is extremely well put together, very well presented and discussed. The subject is well introduced and the methodology employed is very detailed, being extremely easy to follow the authors’ data. The graphism is well done as well, being very easily understood even if without subtitles.

Overall, the manuscript has merit, is scientifically sound and of interest, and deserves publication.

Author Response

Reviewer #2

The manuscript entitled “Concurrent Changes in Extreme Hydroclimate Events in the Colorado River Basin” by Bennett et al. examines concurrent changes in hydroclimate extremes, including heatwaves, droughts, floods, and lowflows, in six historical-to-future (1970-1999, 2070-2099) Earth System Model (ESM) climate scenarios for the Colorado River basin.

The work is extremely well put together, very well presented and discussed. The subject is well introduced and the methodology employed is very detailed, being extremely easy to follow the authors’ data. The graphism is well done as well, being very easily understood even if without subtitles.

Overall, the manuscript has merit, is scientifically sound and of interest, and deserves publication.

Author’s Response to Reviewer #2:

Thank you for the positive comments and review of our work.

Reviewer 3 Report

This is interesting manuscript for publishing at this journal, I recommend some corrections before publishing:

Provide some recommendation in conclusion section

Mention limitation of this study in introduction section

Discussion part should be improved, please compare the results by other researchers’ results.

Figure 5, 6, 7, an 8 require separately explanation.

More literature review is required for Introduction.

Author Response

Reviewer #3

This is interesting manuscript for publishing at this journal, I recommend some corrections before publishing:

Provide some recommendation in conclusion section

Author’s Response to Reviewer #3 Question 1:

Mention limitation of this study in introduction section

Author’s Response to Reviewer #3 Question 2:

We respectfully feel that limitations of the study should be presented in the Discussion section of the paper. We discuss the limitations of the work in the last paragraph of the Discussion section. We have edited this sentence to make it clearer that we are talking about the limitations of the present work.

Discussion part should be improved, please compare the results by other researchers’ results.

Author’s Response to Reviewer #3 Question 3:

We discuss our findings in terms of other work in paragraph six of the study. We could not find any studies that follow our methodology for the CRB. However, we note studies focused on concurrent extreme events, although these studies are using different methods from us. Therefore, we simply discuss other studies that have applied compound events analysis and how they have been used to reveal changes when univariate analysis could not.

Figure 5, 6, 7, an 8 require separately explanation.

Author’s Response to Reviewer #3 Question 4:

We describe figures 5-8 together because we see very the same responses across the time scales, but with different weights associated with them. We have not made any changes to the manuscript.

More literature review is required for Introduction.

Author’s Response to Reviewer #3 Question 5:

We have added a section in the Introduction which reviews the major literature in this subject area, in response to reviewer #1’s question.

Round 2

Reviewer 1 Report

Thank you for providing a new version of the manuscript. Please consider providing a response and marked-up version of the manuscript to facilitate the review process. After carefully comparing this version and the previous version of the manuscripts, the reviewer realized that essential change is very scarce. Therefore, the reviewer’s objections still stand. Below are the comments from the previous round of review. If no significant change is found in the next round of review, the reviewer will reject the publication of this manuscript.

  1. Introduction: From the introduction, the reviewer had the impression that no research has been done before utilizing ESM on this topic. If that is true, please highlight this fast in the manuscript. If not, please add more descriptions of past research utilizing ESM on hydroclimate simulations. The current introduction is missing this element. If necessary, you can merge section 2.2 into the introduction.
  2. Section 2.2: Please provide brief descriptions of the ESMs. What is considered/not considered in the models? What are their differences? What are some of the representative applications in the past?
  3. Section 2.3: Some of the representative input data should be displayed here as graphs, such as temperature and rainfall.
  4. Section 2.4: The application of the “extremes” is not well-explained. For example, how was “heat waves” simulated? Did you use maximum temperature and ET for every day?
  5. Results: 1) Why are there underlines below the “degree” sign? 2) Discussions for Figure 3 can be more comprehensive to cover most variables studies in the figure. 3) Results can be divided and discussed according to the historical and future simulations. This is particularly important since multiple future scenarios were utilized.

Author Response

The current manuscript discusses the hydroclimatic trend in the past and future based on an Earth System Model simulation of the Colorado River Basin. The writing is good and the structure is generally well written. The reviewer had the following comments.

  1. Introduction: From the introduction, the reviewer had the impression that no research has done before utilizing ESM on this topic. If that is true, please highlight this fast in the manuscript. If not, please add more descriptions of past research utilizing ESM on hydroclimate simulations. The current introduction is missing this element. If necessary, you can merge section 2.2 into the introduction.

Author’s Response to Reviewer #1 Question 1:

We have added a section in the introduction that references, in brief, the works that have been carried out utilizing ESMs and other means to examine concurrent extreme hydroclimate events, which is the topic of this paper. Please see below our additional text.

Despite the strong focus on individual indicators, research has emerged that characterizes future changes in extreme events concurrently [1-13]. Review studies on concurrent hydroclimate extremes provide an assessment of all studies and methodological approaches to date of publication [14,15]. A special issue on concurrent or compound extremes was also published [16]. The argument for consideration of concurrent extremes over univariable extreme event analysis is given by several authors, offering definitions, frameworks, paradigms shifts, and generally making the case that examining extreme events in this manner is a means to improve projections of future changes, and that rethinking the traditional univariate approach will allow for physical sciences to be more clearly linked to socioeconomic impacts of extremes [17-21].

Further, we have added a sentence, along with supporting references, on the very general topic of extreme events projection using future climate modeling in the Colorado River basin. Please see below our additional text.

Many studies focused on mean changes in hydroclimate have been carried out for both naturalized and operational aspects of the CRB region [e.g., 22,23-25].

We would like to highlight to the reviewer that the use of ESMs in hydroclimate science is very general, as there are literally thousands of papers on this topic area. We provide references to the major studies throughout the introduction, i.e., references listed below.

Stott, P. How climate change affects extreme weather events. Science 2016, 352, 1517-1518, doi:10.1126/science.aaf7271.

Chen, Y.; Moufouma-Okia, W.; Masson-Delmotte, V.; Zhai, P.; Pirani, A. Recent Progress and Emerging Topics on Weather and Climate Extremes Since the Fifth Assessment Report of the Intergovernmental Panel on Climate Change. Annual Review of Environment and Resources 2018, 43, 35-59, doi:10.1146/annurev-environ-102017-030052.

Trenberth, K.E.; Fasullo, J.T.; Shepherd, T.G. Attribution of climate extreme events. Nature Climate Change 2015, 5, 725-730, doi:10.1038/nclimate2657.

National Academies of Sciences Engineering and Medicine. Attribution of extreme weather events in the context of climate change; National Academies Press: 2016; doi.org/10.17226/21852.

Swain, D.L.; Singh, D.; Touma, D.; Diffenbaugh, N.S. Attributing extreme events to climate change: a new frontier in a warming world. One Earth 2020, 2, 522-527, doi:doi.org/10.1016/j.oneear.2020.05.011.

Field, C.B.; V. Barros; T.F. Stocker; D. Qin; D.J. Dokken; K.L. Ebi; M.D. Mastrandrea; K.J. Mach; G.-K. Plattner; S.K. Allen, et al. Managing the risks of extreme events and disasters to advance climate change adaptation: special report of the intergovernmental panel on climate change; Cambridge University Press: Cambridge, UK and New York, NY, USA, 2012; pp. 582.

Fares, A.; Habibi, H.; Awal, R. Extreme events and climate change: A multidisciplinary approach. In Climate Change and Extreme Events, Elsevier: 2021; pp. 1-7.

Driouech, F.; ElRhaz, K.; Moufouma-Okia, W.; Arjdal, K.; Balhane, S. Assessing future changes of climate extreme events in the CORDEX-MENA region using regional climate model ALADIN-climate. Earth Systems and Environment 2020, 4, 477-492.

  1. Section 2.2: Please provide brief descriptions of the ESMs. What are considered/not considered in the models? What are their differences? What are some of the representative applications in the past?

Author’s Response to Reviewer #1 Question 2:

We have added a sentence to describe the major differences between ESMs, in general, and a reference to a new tool and a recent paper that discusses these differences and some of the history of the global climate models. We have noted that each ESM includes complete representations of the climate system and that they have been widely tested in that regard. The reference provided is an in-depth review on the many aspects of the models and how they differ, including references therein that detail vast number of references on ESMs.

Each ESM differs by a multitude of factors, including but not limited to the country where the model was developed, initial conditions, physics representations, and tuning mechanisms [26]; however, each ESM contains complete representations of the atmosphere, ocean, and land surface and have been widely tested in this regard [27].

  1. Section 2.3: Some of the representative input data should be displayed here as graphs, such as temperature and rainfall.

Author’s Response to Reviewer #1 Question 3:

Historical temperature and precipitation data is provided for each of the ESMs, along with the change, in Table 3 of the manuscript.

  1. Section 2.4: The application of the “extremes” is not well-explained. For example, how were “heat waves” simulated? Did you use maximum temperature and ET for every day?

Author’s Response to Reviewer #1 Question 4:

The description of how we calculated extremes indicators, and the impacts and what they include is provided in Table 1 and Table 2, and in section 2.4 (which includes the details for the peaks over threshold analysis, and the Dn calculations in subsection 2.4.1 and 2.4.2).

For heatwaves, for example, we used maximum temperature and maximum ET for every day (for synoptic), for each of the 12 months, for seasons (each four months period), and for annual. The description is detailed at length over 1.5 pages of the manuscript .

  1. Results: 1) Why are there underlines below the “degree” sign? 2) Discussions for Figure 3 can be more comprehensive to cover most variables studies in the figure. 3) Results can be divided and discussed according to the historical and future simulations. This is particularly important since multiple future scenarios were utilized.

Author’s Response to Question 5.1:

We have replaced the underlined degree symbols with regular degree symbols. Please let us know if we have missed any of these typographic errors.

Author’s Response to Reviewer #1 Question 5.2:

We discuss each of the indicators (tx, evapx, precx, dryd, soilmn, soilmx, qn, qx) in Figure 3 in the first two paragraphs of section 3.1. We added the shorted indicator names to the second paragraph to make that clearer.

Author’s Response to Reviewer #1 Question 5.3:

We chose to write this paper focusing on the Dn statistic (see Section 2.4 of the paper). We feel that this use of this statistic is a novel approach that we have not seen used before. It is also a simple way of explaining the differences between the future and historical data, and it puts all indicators and impacts on the same scale (i.e., normalization as described in the paper) for comparison of the greatest/largest effects. We do not think that adding separate analysis and description of the absolute future and historical results would add much to this paper, and it is considered outside of the scope of the existing manuscript. Indeed, if we were to add this level of detail, we would need to completely redo large sections of our analysis, which is essentially a completely separate paper.

References added to the paper in response to reviewers questions:

  1. Hao, Z.; AghaKouchak, A.; Phillips, T.J. Changes in concurrent monthly precipitation and temperature extremes. Environmental Research Letters 2013, 8, 034014, doi:doi.org/10.1088/1748-9326/8/3/034014.
  2. Mazdiyasni, O.; AghaKouchak, A. Substantial increase in concurrent droughts and heatwaves in the United States. Proceedings of the National Academy of Sciences 2015, 112, 11484-11489, doi:10.1073/pnas.1422945112.
  3. Beniston, M. Trends in joint quantiles of temperature and precipitation in Europe since 1901 and projected for 2100. Geophysical Research Letters 2009, 36, doi:https://doi.org/10.1029/2008GL037119.
  4. Martin, J.-P.; Germain, D. Large-scale teleconnection patterns and synoptic climatology of major snow-avalanche winters in the Presidential Range (New Hampshire, USA). International Journal of Climatology 2017, 37, 109-123, doi:https://doi.org/10.1002/joc.4985.
  5. Wazneh, H.; Arain, M.A.; Coulibaly, P.; Gachon, P. Evaluating the Dependence between Temperature and Precipitation to Better Estimate the Risks of Concurrent Extreme Weather Events. Advances in Meteorology 2020, 2020, 8763631, doi:10.1155/2020/8763631.
  6. Luo, L.; Apps, D.; Arcand, S.; Xu, H.; Pan, M.; Hoerling, M. Contribution of temperature and precipitation anomalies to the California drought during 2012–2015. Geophysical Research Letters 2017, 44, 3184-3192, doi:https://doi.org/10.1002/2016GL072027.
  7. Miao, C.; Sun, Q.; Duan, Q.; Wang, Y. Joint analysis of changes in temperature and precipitation on the Loess Plateau during the period 1961–2011. Climate Dynamics 2016, 47, 3221-3234, doi:10.1007/s00382-016-3022-x.
  8. Fischer, E.M.; Knutti, R. Robust projections of combined humidity and temperature extremes. Nature Climate Change 2013, 3, 126-130, doi:10.1038/nclimate1682.
  9. Estrella, N.; Menzel, A. Recent and future climate extremes arising from changes to the bivariate distribution of temperature and precipitation in Bavaria, Germany. International Journal of Climatology 2013, 33, 1687-1695.
  10. AghaKouchak, A.; Cheng, L.; Mazdiyasni, O.; Farahmand, A. Global warming and changes in risk of concurrent climate extremes: Insights from the 2014 California drought. Geophysical Research Letters 2014, 41, 8847-8852, doi:https://doi.org/10.1002/2014GL062308.
  11. Sedlmeier, K.; Feldmann, H.; Schädler, G. Compound summer temperature and precipitation extremes over central Europe. Theoretical and applied climatology 2018, 131, 1493-1501.
  12. Sedlmeier, K.; Mieruch, S.; Schädler, G.; Kottmeier, C. Compound extremes in a changing climate–a Markov chain approach. Nonlinear Processes in Geophysics 2016, 23, 375-390.
  13. Udall, B.; Overpeck, J. The twenty‐first century Colorado River hot drought and implications for the future. Water Resources Research 2017, 53, 2404-2418.
  14. Hao, Z.; Singh, V.P. Drought characterization from a multivariate perspective: A review. Journal of Hydrology 2015, 527, 668-678, doi:https://doi.org/10.1016/j.jhydrol.2015.05.031.
  15. Hao, Z.; Singh, V.P.; Hao, F. Compound extremes in hydroclimatology: a review. Water 2018, 10, 718.
  16. Beevers, L.; White, C.J.; Pregnolato, M. Editorial to the Special Issue: Impacts of Compound Hydrological Hazards or Extremes. Geosciences 2020, 10, 496.
  17. Diffenbaugh, N.S.; Swain, D.L.; Touma, D. Anthropogenic warming has increased drought risk in California. Proceedings of the National Academy of Sciences 2015, 112, 3931-3936.
  18. Raymond, C.; Horton, R.M.; Zscheischler, J.; Martius, O.; AghaKouchak, A.; Balch, J.; Bowen, S.G.; Camargo, S.J.; Hess, J.; Kornhuber, K., et al. Understanding and managing connected extreme events. Nature Climate Change 2020, 10, 611-621, doi:10.1038/s41558-020-0790-4.
  19. Zscheischler, J.; Westra, S.; van den Hurk, B.J.J.M.; Seneviratne, S.I.; Ward, P.J.; Pitman, A.; AghaKouchak, A.; Bresch, D.N.; Leonard, M.; Wahl, T., et al. Future climate risk from compound events. Nature Climate Change 2018, 8, 469-477, doi:10.1038/s41558-018-0156-3.
  20. Leonard, M.; Westra, S.; Phatak, A.; Lambert, M.; van den Hurk, B.; McInnes, K.; Risbey, J.; Schuster, S.; Jakob, D.; Stafford-Smith, M. A compound event framework for understanding extreme impacts. WIREs Climate Change 2014, 5, 113-128, doi:https://doi.org/10.1002/wcc.252.
  21. Zscheischler, J.; Seneviratne, S.I. Dependence of drivers affects risks associated with compound events. Science advances 2017, 3, e1700263.
  22. Christensen, N.S.; Wood, A.W.; Voisin, N.; Lettenmaier, D.P.; Palmer, R.N. The effects of climate change on the hydrology and water resources of the Colorado River basin. Climatic change 2004, 62, 337-363.
  23. Bennett, K.E.; Bohn, T.J.; Solander, K.; McDowell, N.G.; Vivoni, E.; Middleton, R. Climate-driven disturbances in the San Juan River sub-basin of the Colorado River. Hydrol. Earth Syst. Sci. 2018, 22, 709–725, doi:doi.org/10.5194/hess-22-709-2018.
  24. Bennett, K.E.; Miller, G.; Talsma, C.; Jonko, A.; Bruggeman, A.; Atchley, A.; Lavadie-Bulnes, A.; Kwicklis, E.; Middleton, R. Future water resource shifts in the high desert Southwest of Northern New Mexico, USA. Journal of Hydrology: Regional Studies 2020, 28, 100678, doi:doi.org/10.1016/j.ejrh.2020.100678.
  25. Bennett, K.E.; Tidwell, V.C.; Llewellyn, D.; Behery, S.; Barrett, L.; Stansbury, M.; Middleton, R.S. Threats to a Colorado River Provisioning Basin under Coupled Future Climate and Societal Scenarios. Environmental Research Communications 2019, 095001, doi:https://doi.org/10.1088/2515-7620/ab4028.
  26. Fajardo, J.; Corcoran, D.; Roehrdanz, P.R.; Hannah, L.; Marquet, P.A. GCM compareR: A web application to assess differences and assist in the selection of general circulation models for climate change research. Methods in Ecology and Evolution 2020, 11, 656-663, doi:https://doi.org/10.1111/2041-210X.13360.
  27. Flato, G.; Marotzke, J.; Abiodun, B.; Braconnot, P.; Chou, S.C.; Collins, W.; Cox, P.; Driouech, F.; Emori, S.; Eyring, V. Evaluation of climate models. In Climate change 2013: the physical science basis. Contribution of Working Group I to the Fifth Assessment Report of the Intergovernmental Panel on Climate Change, Cambridge University Press: 2014; pp. 741-866.

Reviewer 3 Report

This manuscript is acceptable.

Author Response

Thank you very much for your review and comments.